**Data Availability Statement:** All relevant data are within the manuscript and its Supporting Information files.

# No bejel among Surinamese, Antillean and Dutch syphilis diagnosed patients in Amsterdam between 2006–2018 evidenced by multi-locus sequence typing of *Treponema pallidum* isolates

**Hélène C. A. Zondag**[1]*, **Sylvia M. Bruisten**[1,2], **Eliška Vrbová**[3], **David Šmajs**[3]

1 Department of Infectious Diseases, Public Health Laboratory, Public Health Service, GGD, Amsterdam, Netherlands, 2 Amsterdam Infection & Immunity Institute, Academic Medical Centre, University of Amsterdam, Amsterdam, Netherlands, 3 Department of Biology, Masaryk University, Brno, Czech Republic

* hzondag@ggd.amsterdam.nl

## Abstract

### Background

*Treponema pallidum* subspecies *pallidum* (TPA) and subsp. *endemicum* (TEN) are the causative agents of syphilis and bejel, respectively. TEN shows similar clinical manifestations and is morphologically and serologically indistinguishable from TPA. Recently, bejel was found outside of its assumed endemic areas. Using molecular typing we aimed to discover bejel and characterize circulating TPA types among syphilis cases with Surinamese, Antillean and Dutch ethnicity in Amsterdam.

### Methods

DNA was extracted from 137 ulcer swabs, which tested positive in the in-house diagnostic PCR targeting the *polA* gene. Samples were collected between 2006 and 2018 from Surinamese, Antillean and Dutch patients attending the Amsterdam STI clinic. Multilocus sequence typing was performed by partial sequence analysis of the *tp0136*, *tp0548* and *tp0705* genes. In addition, the 23S rRNA loci were analyzed for A2058G and A2059G macrolide resistance mutations.

### Results

We found 17 distinct allelic profiles in 103/137 (75%) fully typed samples, which were all TPA and none TEN. Of the strains, 82.5% were SS14-like and 17.5% Nichols-like. The prevalence of Nichols-like strains found in this study is relatively high compared to nearby countries. The most prevalent types were 1.3.1 (42%) and 1.1.1 (19%), in concordance with similar TPA typing studies. The majority of the TPA types found were unique per country. New allelic types (7) and profiles (10) were found. The successfully sequenced 23S rRNA

**Funding:** This work was supported by a grant of the Ministry of Health of the Czech Republic (17-31333A). In addition, H.C.A. Zondag received a Research and Travel grant (FEMS-GO-2018-117) from the federation of European microbiological societies (FEMS; fems-microbiology.org). The funders had no role in study design, data collection and analysis, decision to publish, or preparation of the manuscript.

**Competing interests:** The authors have declared that no competing interests exist.

loci from 123/137 (90%) samples showed the presence of 79% A2058G and 2% A2059G mutations.

## Conclusions

No TEN was found in the samples from different ethnicities residing in Amsterdam, the Netherlands, so no misdiagnoses occurred. Bejel has thus not (yet) spread as a sexually transmitted disease in the Netherlands. The strain diversity found in this study reflects the local male STI clinic population which is a diverse, mixed group.

## Introduction

The spirochetes of the *Treponema* genus consists of different species and subspecies causing syphilis, bejel, yaws and pinta infections. *Treponema pallidum* subspecies *pallidum* (TPA) is the causative pathogen of syphilis, a world-wide prevalent venereal disease. In 2017, there were 33,189 syphilis cases reported in 28 EU/EEA Member States giving an incidence rate of 7.1 cases per 100 000 population [1]. The increasing rates of syphilis cases is mainly driven by behavioral factors and testing strategies by focusing on the risk-group of men who have sex with men (MSM), who accounted for 96% of the 1,224 syphilis cases in the Netherlands in 2018 [2].

Bejel is caused by *T. pallidum* subsp. *endemicum* (TEN) and was, until recently, thought to be non-venereal [3–5]. Bejel shows similar clinical manifestations and is morphologically and serologically indistinguishable from TPA [6]. In 2016, a TEN isolate was identified in France [7]. A recently developed multilocus sequence typing (MLST) scheme [8] enables differentiation between treponemal subspecies as well as the distinction between the two major genetic clades within TPA, Nichols and SS14, and provides strain types within these clades. In 2018, this MLST method was used in Cuba [6] and Japan [9] to retrospectively investigate treponemal subspecies in samples of patients that were diagnosed with syphilis. Interestingly, both studies found cases of bejel, caused by TEN, which strongly suggested sexual transmission of this disease and showed bejel cases outside of the known endemic areas, Sahelian Africa and Saudi Arabia [4].

This study aimed to discover TEN strains causing bejel among syphilis cases from patients with Surinamese or Antillean ethnicity assuming possible importation of bejel from their country of origin. We hypothesized that if bejel was also misdiagnosed in Amsterdam, as was the case in Cuba among patients with syphilis [6], we would more likely find TEN in Dutch patients with a Surinamese or Antillean ethnicity, as these countries are geographically close to Cuba (Fig 1), than in patients with a Dutch ethnicity. Dutch patients were also included to investigate the presence of bejel in Amsterdam.

In addition, molecular characterization data was used to increase the epidemiological knowledge of the strain types and investigate possible associations between allelic types, profiles and patient's clinical and demographic data.

## Methods

### Sample selection and preparation

Based on a positive *polA* PCR on genital ulcer swabs [10] 137 samples were retrospectively selected from patients with a Surinamese, Antillean or Dutch nationality visiting the STI clinic

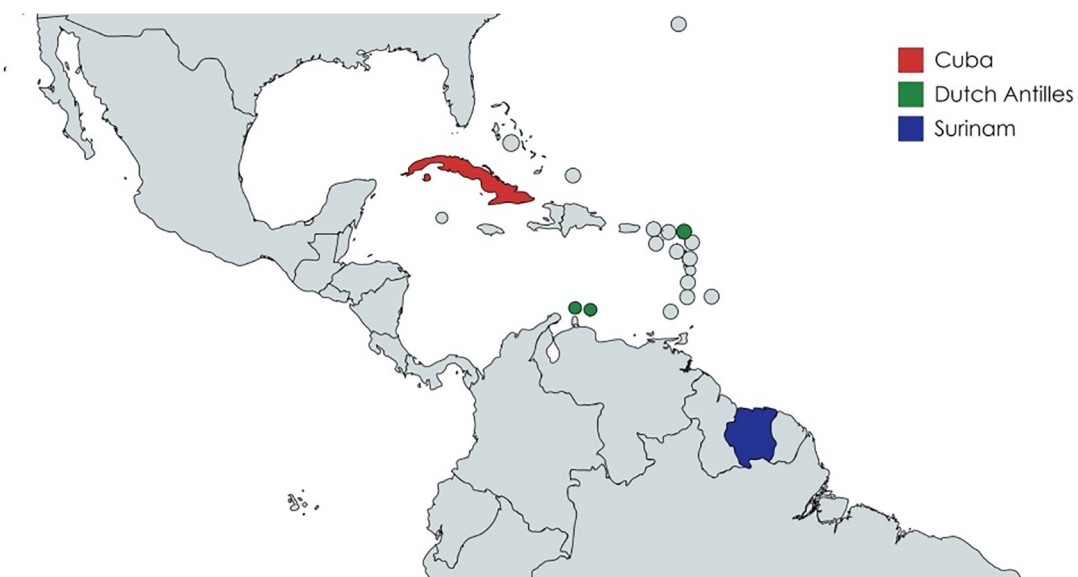

**Fig 1. Map showing geographical location of Cuba (red), the Dutch Antilles (green) and Surinam (blue).** This map was constructed using matchart.net.

in Amsterdam between 2006 and 2018. Within this time frame all available samples from syphilis diagnosed Surinamese and Antillean patients were included in the study. For a more representative and equal spread of samples the number of Dutch samples were randomly selected per calendar year to match the number of samples from Antillean and Surinamese patients.

If the volume of the DNA isolate was insufficient, DNA from the original patient sample, which was stored at -20˚C, was re-extracted using isopropanol precipitation method [11]. Demographic and clinical data was collected for all patients.

## Defining clinical stages of syphilis

The primary stage of syphilis is defined as an oro-, ano-genital ulcerative disease with a positive dark field microscopy and/or with a positive ulcer swab PCR result. Criteria for secondary syphilis are a rash with or without lymphadenopathy, or mucosal lesions such as condylomata lata, and an RPR $\geq$ 1:4. Ulcers may also occur in patients presenting with a rash with or without lymphadenopathy, or mucosal lesions. These patients are diagnosed with secondary syphilis. All samples from this study are ulcer swabs and were derived from both (primary and secondary) clinical syphilis stages.

## Serological testing

Serologically, until 2013 a *Treponema pallidum* particle agglutination (TPPA) assay was performed on all samples. After 2013, the enzyme immunoassay (EIA) for the detection of IgM antibodies to *Treponema pallidum* was introduced and used on all samples unless the patient had a syphilis infection before. In addition, quantitative rapid plasma reagin (RPR) flocculation test (RPR-Nosticon II; bioMérieux) was performed according to the specifications of the manufacturers.

## Molecular typing method

Molecular characterization of the samples was performed using the recently developed MLST method by Grillová et al. 2018 [8]. This MLST method is based on the partial amplification and sequence analysis of three chromosomal loci; *tp0136*, *tp0548* and *tp0705*. Also both 23S rRNA genes were partly sequenced to check for the A2058G and A2059G mutations associated with macrolide resistance. The partial amplification and sequence analysis using Sanger sequencing was performed as described [8, 12]. Sequence analyses were performed using Bionumerics version 7.6.3 (Applied Maths, BioMérieux). All allelic variants and allelic profiles were added to the PubMLST BIGSdb database of TPA [13]. New allelic variants and profiles were given subsequent numbers adding to the TPA database. Typed isolates were uploaded when 2 or more allelic variants were identified.

## Genetic clade and subspecies distinction

Clade determination (SS14-like or Nichols-like) was possible with the partly sequenced regions of *tp0136* and *tp0548*. Every new allelic variant was compared to both references in order to determine its genetic group. To visualize subspecies and genetic clades a phylogenetic tree of the concatenated sequences was generated with the bootstrapping maximum-likelihood algorithm and the Tamura-Nei method in MEGA6.06 [14].

## Data analysis

Allelic profiles, demographic and clinical data were tested for possible associations with Fisher's Exact Tests and, where possible, Pearson's Chi-square Tests between patients with fully typable and not (fully) typable samples using IBM SPSS Statistics (version 21.0.0.2). A $p < 0.05$ was considered significant.

## Ethical clearing

This study was reviewed, and the need for consent was waived by our Institutional Review Board, which is the Medical Ethical Committee of the Amsterdam University Medical Centers in the Netherlands. According to the Dutch Medical Research Act Involving Human Subjects on use of retrospective diagnostic material no additional ethical approval was required for this study (W19_113#19.146). An opt-out system is used at the Public Health Service of Amsterdam to assure that if patients object to having their samples used for research that these are destroyed. Only anonymized patient data were used as provided by an independent datamanager. No samples from patients under the age of 18 years old were included.

## Results

Isolated DNA samples were available from 137 ulcer swabs and derived from 24 Antillean patients, 46 Surinamese patients and 67 Dutch patients. Patient characteristics were collected for all 137 patients and are shown in Table 1. Dutch patients had a higher median age, 46 years with an interquartile range (IQR) of 38–51, compared to 37 years (IQR 29–45) in Surinamese and 35 years (IQR 31–41) in Antillean patients. Only 67% of the Surinamese patients were MSM based on their sexual behavior in the past 6 months. This is much lower compared to the 91% and 92% among the Dutch and Antillean patients. The HIV status among all ethnicities was similar with 40% HIV positive Dutch patients, 39% Surinamese and 46% Antillean.

From the 137 DNA samples 103 (75%) were successfully amplified and analyzed for all typing loci. This resulted in 17 distinct allelic profiles (Fig 2). Of these samples, 85 (82.5%) were

**Table 1. Demographic and clinical characteristics of all 137 patients.**

| Clinical characteristics of patients | (n = 137) |
|---|---|
| **General** | |
| Ethnicity | |
| Antillean | 24 |
| Surinamese | 46 |
| Dutch | 67 |
| Median age (IQR) | 41 (34–48) |
| Sex (%) | |
| Male | 136 (99.3) |
| Female | 1 (0.7) |
| Sexual behavior (%) | |
| MSM | 114 (83.2) |
| MWMW | 11 (8.0) |
| MSW | 11 (8.0) |
| WSM | 1 (0.7) |
| HIV status (%) | |
| Positive | 56 (40.9) |
| Negative | 75 (54.7) |
| Unknown | 6 (4.4) |
| **Serology** | |
| RPR (%) | |
| High (1:32 ≤) | 43 (31.4) |
| Middle (1:4–1:16) | 42 (30.7) |
| Low (1:1–1:2) | 26 (19.0) |
| Negative | 24 (17.5) |
| Unknown | 2 (1.5) |
| TPPA/EIA (%) | |
| Positive | 102 (74.5) |
| Negative | 4 (2.9) |
| Not tested | 26 (18.9) |
| Unknown | 5 (3.7) |
| **Syphilis stage** | |
| Primary syphilis (%) | 108 (78.8) |
| Secondary syphilis (%) | 29 (21.2) |

MSM, men who have sex with men; MSMW, men who have sex with men and women; MSW, men who have sex with women; WSM, women who have sex with men.

SS14-like and 18 (17.5%) Nichols-like strains. The most common allelic profile found was 1.3.1 occurring in 42/99 (42%) isolates. None of the samples in this study were TEN.

In addition, a total of 6 allelic variants were found for locus *tp0136*, 11 for *tp0548* and 5 for *tp0705*. Of these, 7 were new allelic variants, 2 for the *tp0136* locus (numbers 19 and 20) and 5 for the *tp0548* locus (numbers 43–47). In total, 10 new allelic profiles were found with new and known allelic variants giving rise to a total of 10 new ST (numbered 56–65, Table 2).

All allelic profiles found in this study were visualized in a phylogenetic tree using the concatenated sequences (Fig 3). Their genetic diversity within the two major genetic clades, SS14 and Nichols, is clearly shown with 13 SS14-like allelic profiles and 4 Nichols-like allelic profiles.

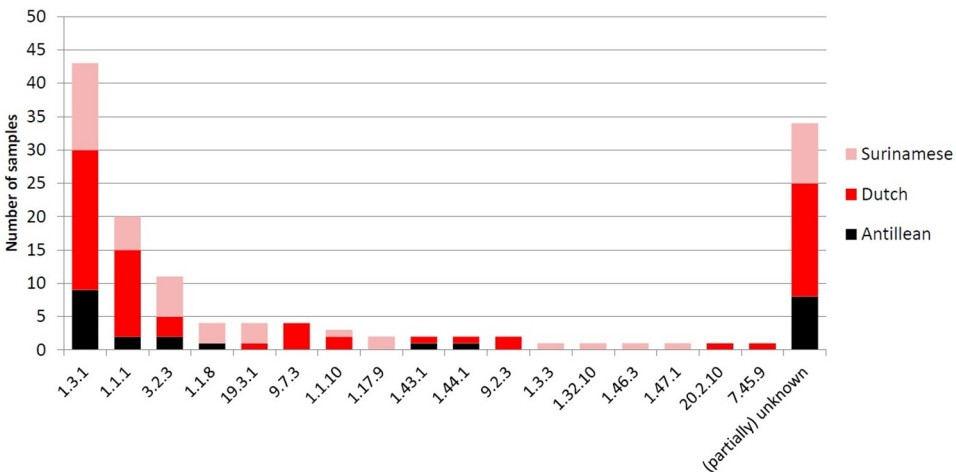

**Fig 2. An overview of allelic profiles colored by ethnicity.** Molecular typing of 137 Treponema pallidum subspecies pallidum isolates from Amsterdam.

There was a significant difference in the typability of ulcer swab samples from patients with primary syphilis, of which the isolates were more often fully typed, as compared to patients with secondary syphilis (S1c Table). In addition, secondary syphilis patients have a significantly higher RPR titer than primary patients (unpublished data) and all fully typed secondary syphilis isolates (n = 18) contained SS14-like TPA strains (S2 Table). No associations were found between the TPA types and ethnicity (Fig 2). Also, no significant differences were found based on the patient's HIV status, syphilitic stage or RPR titer of the typable isolates versus the non-typable isolates (S1 Table).

**Table 2. Allelic profiles identified from the 103 fully typed samples in this study.**

| Sequence type | Allelic profile | 23S rDNA (no. of samples) | Genetic group | Frequency |
|---|---|---|---|---|
| 1 | 1.3.1 | R8(41)/X(2) | SS14-like | 43 |
| 2 | 1.1.1 | S(11)/R8(7)/R9(2) | SS14-like | 20 |
| 6 | 3.2.3 | R(11) | Nichols-like | 11 |
| 3 | 1.1.8 | S(2)/R(2) | SS14-like | 4 |
| 56* | 19*.3.1 | R8(4) | SS14-like | 4 |
| 26 | 9.7.3 | S(1)/R8(3) | Nichols-like | 4 |
| 19 | 1.1.10 | S(3) | SS14-like | 3 |
| 28 | 1.17.9 | R8(2) | SS14-like | 2 |
| 57* | 1.43*.1 | R8(2) | SS14-like | 2 |
| 58* | 1.44*.1 | R8(2) | SS14-like | 2 |
| 60* | 9.2.3 | S(2) | Nichols-like | 2 |
| 61* | 1.3.3 | R8(1) | SS14-like | 1 |
| 62* | 1.32.10 | S(1) | SS14-like | 1 |
| 63* | 1.46*.3 | S(1) | SS14-like | 1 |
| 64* | 1.47*.1 | R8(1) | SS14-like | 1 |
| 65* | 20*.2.10 | S(1) | Nichols-like | 1 |
| 59* | 7.45*.9 | R8(1) | SS14-like | 1 |

*New sequence types and allelic variants were added to the BIGSdb database for *Treponema pallidum* subspecies *pallidum* [13].

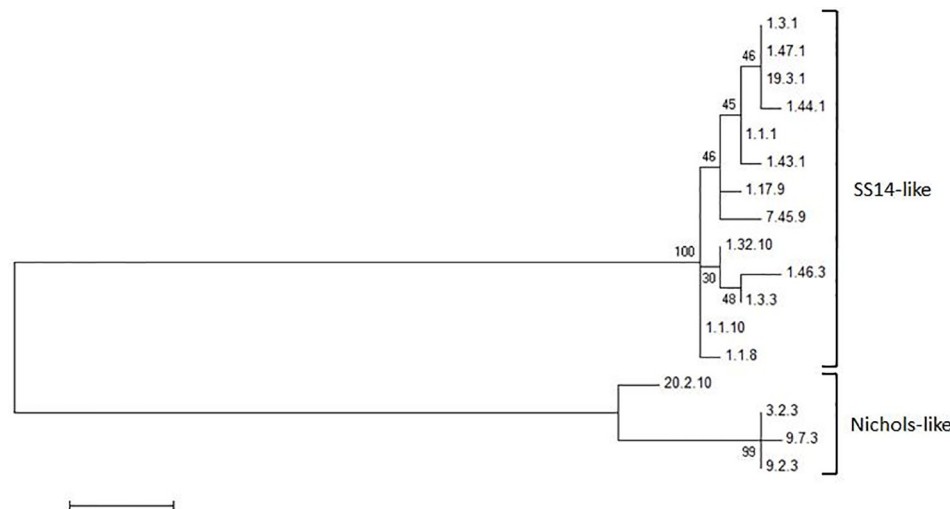

**Fig 3. Phylogenic tree (unrooted) of the 17 allelic profiles.** The Maximum Likelihood tree was constructed using the Timura-Nei method in MEGA6.06 with 1000 bootstraps.

There were 123/137 (90%) isolates successfully sequenced for the relevant parts of the 23S rRNA genes and 81% of all isolates contained one of the macrolide resistance mutations, 79% contained the A2058G mutation and 2% the A2059G mutation. None of the isolates carried both mutations. Both of the samples containing the A2059G mutation had allelic profile 1.1.1. The prevalence of macrolide resistance causing mutations in the 23S rRNA genes showed an increased trend over time from 53% in 2007 to 79% in 2017 (Fig 4).

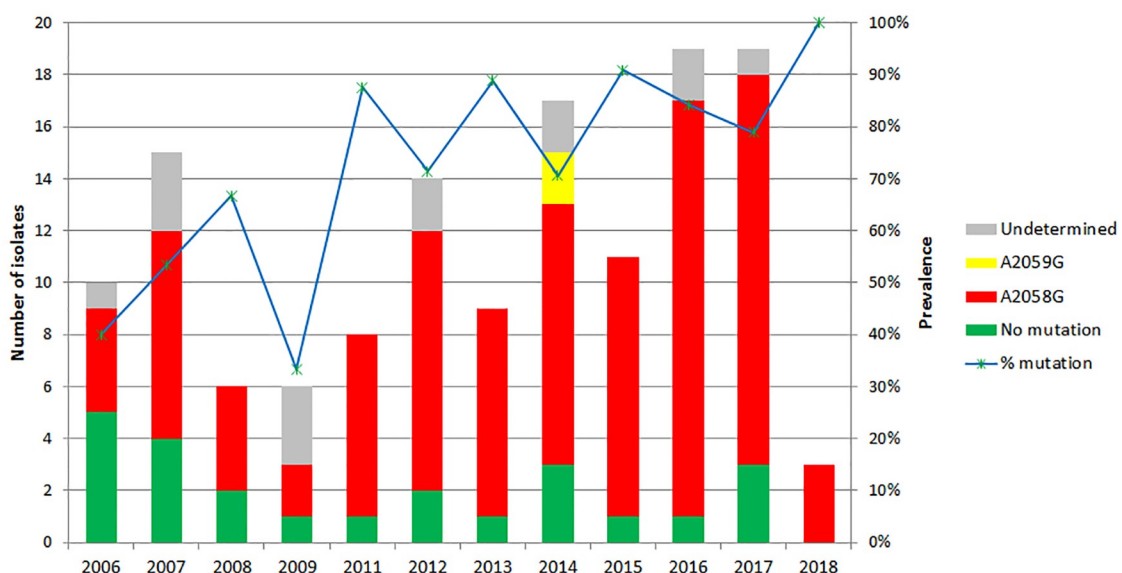

**Fig 4. Macrolide resistance causing mutations over time among the included isolates in Amsterdam between 2006 and 2018.**

## Discussion

No TEN was found suggesting that bejel has not (yet) spread as a sexually transmitted disease to the Netherlands. By including Surinamese and Antillean patients who visited the Amsterdam STI clinic we aimed to increase the likelihood of finding strains from that geographical location which is close to Cuba where bejel was found within syphilis diagnosed patients [6]. An important limitation of this study is that we only had samples available from persons who reside in the Netherlands, probably all in the Amsterdam region. In addition, no data was available on the location of the acquisition of infection nor the ethnicity of the partners of these patients. Among these patients in Amsterdam with an Antillean or Surinamese ethnicity we did not see TEN infections, but it is still possible that bejel occurs in their countries of origin.

The most prevalent allelic profile found in this study was the SS14-like strain, 1.3.1 (42%), followed by 1.1.1 (19%) in concordance with similar TPA MLST studies [8, 12, 15, 16]. Until now, these two types have been found in every TPA MLST study [13], while all geographic locations (Czech Republic [16], Switzerland [8], France [12], Cuba [15], and the Netherlands) from which the samples were derived also have less common and unique allelic profiles suggesting the combination of global mixing populations and more local mixing of the hosts (S3 Table). To investigate possible bias in the typable versus not (fully) typable samples these groups were compared based on ethnicity, RPR titer, HIV status and syphilis stage. Of these variables, the syphilis stage of the patient showed a significant effect on the typability of the sample (S1c Table). Ulcer swab samples from patients with primary stage syphilis were significantly more likely to be fully typed than samples from secondary stage syphilis patients. A possible explanation may be that ulcers from primary syphilis patients contain more serous fluid as compared to the mostly dried and older ulcers present in secondary syphilis patients. Not surprisingly, patients with a secondary syphilis infection have a significant higher RPR titer than primary syphilis patients.

The use of the recently developed MLST for TPA strains allowed molecular characterization and subspecies determination with a high resolution for 103/137 (75%) of the selected samples. Obtaining a full type was more challenging for the older samples. Similar studies using this sequence based typing method obtained full types for up to 94% [15]. A previous molecular characterization study using the enhanced CDC method on isolates from Amsterdam showed a similar percentage of fully typed samples and a comparable typing distribution [17]. However, the enhanced CDC method combines partial sequence analysis of *tp0548* with the analysis of a restriction fragment length pattern of *tpr* genes and the number of acidic repeat proteins of 60bp each making it a complicated and occasionally unstable method [17] for TPA typing.

The genetic clade distribution found in this study were 83% SS14-like strains and 17% Nichols-like strains. This Nichols-like prevalence is more comparable to Argentina, Peru and Taiwan than to the countries closer to the Netherlands like France, Denmark, Ireland, the UK and the Czech Republic [18]. Worldwide only 117/1989 (5.9%) clinical samples were classified as Nichols-like [19]. This relatively high ratio of Nichols-like strains versus SS14-like strains was not explained by ethnicity as 2/16 (13%) fully typed samples from Antillean patients, 6/37 (16%) samples from Surinamese patients and 10/50 (20%) samples from Dutch patients, contained Nichols-like strains. Interestingly, all fully typed isolates from secondary stage syphilis patients contained TPA strains belonging to the SS14 clade (S2 Table), whereas an association was found between secondary stage syphilis and Nichols-like strains in a previous study [12].

Seven new allelic variants and 10 new ST were found adding to the knowledge of TPA strain diversity. All allelic variants and ST were added to the pubMLST BIGSdb database of TPA which was recently published for the surveillance and epidemiology of syphilis [13].

Furthermore, the successfully sequenced part of the 23S rRNA genes from 123/137 (90%) samples showed the presence of A2058G and A2059G mutations, 79% and 2% respectively. When analyzing the samples over time an increase in macrolide resistant mutations was seen from 53% in 2007 to 79% in 2017. The samples were not selected to investigate this as the distribution is not ideal, but the upward trend is significant and supports findings in earlier studies focusing on this topic [20].

The strain diversity found in this study reflects the local male STI clinic population which is a diverse, mixed group. Future studies should collect samples from the specific country of interest as ethnicity is not enough to investigate the epidemiology of bejel. Molecular characterization of the TPA bacteria remains important for network analyses and uncovering pathogenic associations with certain genetic variants.

## Supporting information

**S1 Table. Fisher's Exact and Pearson Chi-squared tests for ethnicity, RPR titer, syphilis stage and HIV status versus typability of the samples.** S1a) Table. Pearson Chi-squared test for ethnicity and typability of samples. S1b) Table. Fisher's Exact test for RPR titer and typability of samples. S1c) Table. Pearson Chi-squared test for syphilis stage and typability of samples. S1d) Table. Fisher's Exact test for HIV status and typability of samples.
(DOCX)

**S2 Table. Fisher's Exact test for syphilis stage versus genetic *Treponema pallidum* subspecies *pallidum* clade.**
(DOCX)

**S3 Table. Overview of full MLST *Treponema pallidum* subspecies *pallidum* types found in the public database [13].**
(DOCX)

## Author Contributions

**Conceptualization:** Hélène C. A. Zondag, Sylvia M. Bruisten, David Šmajs.

**Data curation:** Hélène C. A. Zondag.

**Formal analysis:** Hélène C. A. Zondag.

**Funding acquisition:** Hélène C. A. Zondag, Sylvia M. Bruisten, David Šmajs.

**Investigation:** Hélène C. A. Zondag, Eliška Vrbová.

**Methodology:** Eliška Vrbová, David Šmajs.

**Resources:** David Šmajs.

**Supervision:** Sylvia M. Bruisten, David Šmajs.

**Writing – original draft:** Hélène C. A. Zondag.

**Writing – review & editing:** Hélène C. A. Zondag, Sylvia M. Bruisten, Eliška Vrbová, David Šmajs.

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
