## [Decision Letter · Decision Letter 0]

4 Feb 2020

PONE-D-19-35022

No bejel among Surinamese, Antillean and Dutch syphilis diagnosed patients in Amsterdam between 2006 – 2018 evidenced by multi-locus sequencing of *Treponema pallidum*isolates

PLOS ONE

Dear Dr Zondag,

Thank you for submitting your manuscript to PLOS ONE. After careful consideration, we feel that it has merit but does not fully meet PLOS ONE’s publication criteria as it currently stands. Therefore, we invite you to submit a revised version of the manuscript that addresses the points raised during the review process.

In addition to the first reviewer's request to provide more details on the samples obtained from secondary syphilis patients, we ask you to clarify under the method section how you defined primary and secondary syphilis. 

We would appreciate receiving your revised manuscript by the 1st of March 2020. To enhance the reproducibility of your results, we recommend that if applicable you deposit your laboratory protocols in protocols.io, where a protocol can be assigned its own identifier (DOI) such that it can be cited independently in the future. For instructions see: http://journals.plos.org/plosone/s/submission-guidelines#loc-laboratory-protocols

We look forward to receiving your revised manuscript.

Kind regards,

Tania Crucitti

Academic Editor

PLOS ONE

Journal Requirements:

1. Please provide additional details regarding participant consent. In the ethics statement in the Methods and online submission information, please ensure that you have specified (1) whether consent was informed and (2) what type you obtained (for instance, written or verbal, and if verbal, how it was documented and witnessed). If your study included minors, state whether you obtained consent from parents or guardians. If the need for consent was waived by the ethics committee, please include this information.

Reviewers' comments:

Reviewer's Responses to Questions

**Comments to the Author**

1. Is the manuscript technically sound, and do the data support the conclusions?

Reviewer #1: Partly

Reviewer #2: Yes

2. Has the statistical analysis been performed appropriately and rigorously? 

Reviewer #1: N/A

Reviewer #2: Yes

3. Have the authors made all data underlying the findings in their manuscript fully available?

Reviewer #1: Yes

Reviewer #2: Yes

4. Is the manuscript presented in an intelligible fashion and written in standard English?

Reviewer #1: Yes

Reviewer #2: Yes

5. Review Comments to the Author

Reviewer #1: In this manuscript by Zondag et al the possibility that TEN had reached the Nederlands is evaluated by typing Tp strains in patients with three different ethnicity attending STD clinics in the Netherlands.

Although the analysis of the samples obtained is technically sound and the results of the molecular analysis are clear and interesting, this reviewer does not particularity agree with the assumption that just looking at strains from patients with Antillean or Surinamese ethnicity is a adequate experimental design to address the question the authors pose.

Clearly the authors believe that these non-Dutch patients could have imported TEN case form their home countries, but in reality no data are provided to support that these infections were acquired elsewhere and not in the Nederlands. One should safely assume that these patients were not tourists, but no data are given about how long they lived in the Netherlands, ethnicity of their sexual partners and so on. So, in the context given, differentiating based on ethnicity seems rather nonscientific. Either the author revise their assumptions, or provide better justification for looking at such ethnic groups. Similarly, this reviewer believes that the authors' conclusion that "TEN has not (yet) reached the Netherlands does not have solid scientific basis, because it seems that 1) there are no data on TEN presence in Suriname or Antilles, and 2) with these samples, the authors might not have looked at all outside the local Dutch sexual network.

More detail should be given about the kind of samples obtained for analysis from secondary syphilis patients.

Reviewer #2: THis is a well written paper and I have no major concerns. THe finding is probably not that surprising and as the authors acknowledge it does not preclude bejel being transmitted in the countries of origin. I have no major comments requiring changes.

6. PLOS authors have the option to publish the peer review history of their article (what does this mean?). If published, this will include your full peer review and any attached files.

Reviewer #1: No

Reviewer #2: No

---

## [Author Response · Author response to Decision Letter 0]

21 Feb 2020

To the Academic Editor

Prof. Tania Crucitti

PLOS ONE

Amsterdam, 21 February 2020

Subject: Revised manuscript PONE-D-19-35022

Dear Prof. Tania Crucitti, dear Reviewers,

On behalf of all authors of manuscript PONE-D-19-35022 ("No bejel among Surinamese, Antillean and Dutch syphilis diagnosed patients in Amsterdam between 2006 – 2018 evidenced by multi-locus sequence typing of Treponema pallidum isolates"), we thank the Academic Editor and Reviewers for their careful reading and constructive comments. We have addressed the comments of the Academic Editor and Reviewers and revised the manuscript according to their suggestions. We have submitted a new version of the manuscript; the parts we changed according to the Academic Editor’s and Reviewer’s comments are shown in the ‘track changes’. Below we have addressed the comments point by point in more detail.

We declare that all authors have seen and approved the manuscript, and have contributed significantly to the work. The manuscript presented here has not been published and is not being considered for publication elsewhere. None of the authors have declared a conflict of interest. 

Finally, we did not receive any writing assistance other than copy-editing in the preparation of the manuscript. 

We sincerely hope that you will consider our manuscript for publication in PLOS ONE.

On behalf of all co-authors,

Yours sincerely,

Hélène Zondag (corresponding author)

Public Health Laboratory, 

Public Health Service of Amsterdam (GGD Amsterdam), 

Nieuwe Achtergracht 100, 

1018 WT Amsterdam, 

the Netherlands

Phone: +31-20-5559205

E-mail: hzondag@ggd.amsterdam.nl

EDITOR’S COMMENTS 

1. We ask you to clarify under the method section how you defined primary and secondary syphilis.

Response: Thank you for your suggestion. We added the subsection “Defining clinical stages of syphilis” in the method section regarding the definition of the clinical syphilitic stages; “The primary stage of syphilis is defined as an oro-, ano-genital ulcerative disease with a positive dark field microscopy and/or with a positive ulcer swab PCR result. Criteria for secondary syphilis are a rash with or without lymphadenopathy, or mucosal lesions such as condylomata lata, and an RPR ≥ 1:4. Ulcers may also occur in patients presenting with a rash with or without lymphadenopathy, or mucosal lesions. These patients are diagnosed with secondary syphilis. All samples from this study are ulcer swabs and were derived from both (primary and secondary) clinical syphilis stages.”

Response: All files (and filenames) have now been formatted to meet PLOS ONE’s style requirements. 

Response: To increase transparency and to clarify our ethical clearing we added the requested details including the waived need for consent and the opt-out system regarding use of rest-materials from diagnostic samples (lines 153 – 160). No minors were included. “This study was reviewed and the need for consent was waived by our Institutional Review Board, which is the Medical Ethical Committee of the Amsterdam University Medical Centers in the Netherlands. According to the Dutch Medical Research Act Involving Human Subjects on use of retrospective diagnostic material no additional ethical approval was required for this study (W19_113#19.146). An opt-out system is used at the Public Health Service of Amsterdam to assure that if patients object to having their samples used for research that these are destroyed. Only anonymized patient data were used as provided by an independent datamanager. No samples from patients under the age of 18 years old were included.”

REVIEWERS' COMMENTS

Reviewer: 1

1. In this manuscript by Zondag et al the possibility that TEN had reached the Nederlands is evaluated by typing Tp strains in patients with three different ethnicity attending STD clinics in the Netherlands.

Although the analysis of the samples obtained is technically sound and the results of the molecular analysis are clear and interesting, this reviewer does not particularity agree with the assumption that just looking at strains from patients with Antillean or Surinamese ethnicity is a adequate experimental design to address the question the authors pose.

Clearly the authors believe that these non-Dutch patients could have imported TEN case form their home countries, but in reality no data are provided to support that these infections were acquired elsewhere and not in the Nederlands. One should safely assume that these patients were not tourists, but no data are given about how long they lived in the Netherlands, ethnicity of their sexual partners and so on. So, in the context given, differentiating based on ethnicity seems rather nonscientific. Either the author revise their assumptions, or provide better justification for looking at such ethnic groups. 

Response: Thank you for your critical reading and comments. We regret that our objectives were not stated clearly and have now made an attempt to clarify these aims better. 

We were interested in discovering whether we could find bejel among our patients in the Amsterdam region who were diagnosed with syphilis. Lines 83-89 state our hypothesis for choosing the ethnicities in this study. We have rephrased this paragraph to more clearly describe and explain the study. “This study aimed to discover TEN strains causing bejel among syphilis cases from patients with Surinamese or Antillean ethnicity assuming possible importation of bejel from their country of origin. We hypothesized that if bejel was also misdiagnosed in Amsterdam, as was the case in Cuba among patients with syphilis [6], we would more likely find TEN in Dutch patients with a Surinamese or Antillean ethnicity, as these countries are geographically close to Cuba (Fig 1), than in patients with a Dutch ethnicity. Dutch patients were also included to investigate the presence of bejel in Amsterdam.” We hypothesized that if bejel was misdiagnosed in Amsterdam, as was the case in Cuba among patients with syphilis (Noda et al, 2018) we maybe had a higher chance of detecting this phenomenon in patients whose roots were in countries closer to Cuba and who are Dutch citizens. Therefore we included patients from Surinam or the Antillean Islands since these countries are geographically close to Cuba. 

Indeed, as the reviewer correctly assumes, these patients were not tourists, but actually Dutch citizens, which is what we stated in line 225 “we only had samples available from persons who reside in the Netherlands, probably all in the Amsterdam region”. 

Unfortunately no additional data is available on the location of the acquisition of the infection or on the ethnicity of the partners of these patients. To emphasize these limitations in the Discussion section we rephrased and expanded the following sentence (lines 224-229) “An important limitation of this study is that we only had samples available from persons who reside in the Netherlands, probably all in the Amsterdam region. In addition, no data was available on the location of the acquisition of infection nor on the ethnicity of the partners of these patients. Among these patients in Amsterdam with an Antillean or Surinamese ethnicity we did not see TEN infections, but it is still possible that bejel occurs in their countries of origin.” 

2. Similarly, this reviewer believes that the authors' conclusion that "TEN has not (yet) reached the Netherlands does not have solid scientific basis, because it seems that 1) there are no data on TEN presence in Suriname or Antilles, and 2) with these samples, the authors might not have looked at all outside the local Dutch sexual network.

Response: We agree with the reviewer that it would indeed have been very interesting to also have samples from patients who live in Suriname and/or the Antilles, but we did not have these samples. See also our response to comment 1 why we still choose the samples as we did.

We also acknowledged the fact that these patients are residents from the Amsterdam region in line 225 “we only had samples available from persons who reside in the Netherlands, probably all in the Amsterdam region.” With this in mind, stating the finding in our case like in line 53 (Abstract) “Bejel has thus not (yet) spread as a sexually transmitted disease in the Netherlands.“ and line 221 “No TEN was found suggesting that bejel has not (yet) spread as a sexually transmitted disease to the Netherlands.” is a scientifically correct statement to make. 

3. More detail should be given about the kind of samples obtained for analysis from secondary syphilis patients.

Response: Thank you for your suggestion. We added a subsection under Methods named “Defining clinical stages of syphilis”. By adding the following sentence (lines 119 – 120) “All samples from this study are ulcer swabs and were derived from both (primary and secondary) clinical syphilis stages.” we hope to have given more clarity on the type of samples used from secondary syphilis patients. Please also see our answer to the editor concerning the definition of primary and secondary syphilis cases.

Reviewer: 2

This is a well written paper and I have no major concerns. The finding is probably not that surprising and as the authors acknowledge it does not preclude bejel being transmitted in the countries of origin. I have no major comments requiring changes.

Response: Thank you for your positive response.

---

## [Decision Letter · Decision Letter 1]

26 Feb 2020

No bejel among Surinamese, Antillean and Dutch syphilis diagnosed patients in Amsterdam between 2006 – 2018 evidenced by multi-locus sequence typing of *Treponema pallidum* isolates

PONE-D-19-35022R1

Dear Dr. Zondag,

We are pleased to inform you that your manuscript has been judged scientifically suitable for publication and will be formally accepted for publication once it complies with all outstanding technical requirements.

With kind regards,

Tania Crucitti

Academic Editor

PLOS ONE

Additional Editor Comments (optional):

Reviewers' comments:

Reviewer's Responses to Questions

**Comments to the Author**

1. If the authors have adequately addressed your comments raised in a previous round of review and you feel that this manuscript is now acceptable for publication, you may indicate that here to bypass the “Comments to the Author” section, enter your conflict of interest statement in the “Confidential to Editor” section, and submit your "Accept" recommendation.

Reviewer #1: All comments have been addressed

2. Is the manuscript technically sound, and do the data support the conclusions?

Reviewer #1: Yes

3. Has the statistical analysis been performed appropriately and rigorously? 

Reviewer #1: Yes

4. Have the authors made all data underlying the findings in their manuscript fully available?

Reviewer #1: Yes

5. Is the manuscript presented in an intelligible fashion and written in standard English?

Reviewer #1: Yes

6. Review Comments to the Author

Reviewer #1: The authors have addressed this reviewer's comments satisfactorily. I suggest acceptance of the manuscript

7. PLOS authors have the option to publish the peer review history of their article (what does this mean?). If published, this will include your full peer review and any attached files.

Reviewer #1: No

---

## [Editor Report · Acceptance letter]

28 Feb 2020

PONE-D-19-35022R1 

No bejel among Surinamese, Antillean and Dutch syphilis diagnosed patients in Amsterdam between 2006 – 2018 evidenced by multi-locus sequence typing of *Treponema pallidum* isolates 

Dear Dr. Zondag:

I am pleased to inform you that your manuscript has been deemed suitable for publication in PLOS ONE. Congratulations! Your manuscript is now with our production department. 

With kind regards,

on behalf of

Dr. Tania Crucitti 

Academic Editor

PLOS ONE